# Stem Cell-Derived Extracellular Vesicles and Kidney Regeneration

**DOI:** 10.3390/cells8101240

**Published:** 2019-10-11

**Authors:** Cristina Grange, Renata Skovronova, Federica Marabese, Benedetta Bussolati

**Affiliations:** 1Department of Medical Sciences, University of Turin, via Nizza 52, 10126 Torino, Italy; cristina.grange@unito.it; 2Department of Molecular Biotechnology and Health Sciences, University of Turin, via Nizza 52, 10126 Torino, Italy; renata.skovronova@unito.it (R.S.); federica.marabese@edu.unito.it (F.M.); 3Maria Pia Hospital, 10132 Turin, Italy

**Keywords:** AKI, CKD, exosomes, regenerative medicine, renal injury

## Abstract

Extracellular vesicles (EVs) are membranous vesicles containing active proteins, lipids, and different types of genetic material such as miRNAs, mRNAs, and DNAs related to the characteristics of the originating cell. They possess a distinctive capacity to communicate over long distances. EVs have been involved in the modulation of several pathophysiological conditions and, more importantly, stem cell-derived EVs appear as a new promising therapeutic option. In fact, several reports provide convincing evidence of the regenerative potential of EVs released by stem cells and, in particular, mesenchymal stromal cells (MSCs) in different kidney injury models. Described mechanisms involve the reprogramming of injured cells, cell proliferation and angiogenesis, and inhibition of cell apoptosis and inflammation. Besides, the therapeutic use of MSC-EVs in clinical trials is under investigation. This review will focus on MSC-EV applications in preclinical models of acute and chronic renal damage including recent data on their use in kidney transplant conditioning. Moreover, ongoing clinical trials are described. Finally, new strategies to broaden and enhance EV therapeutic efficacy by engineering are discussed.

## 1. Introduction

Renal failure is one of the most significant causes of mortality and morbidity all over the world [1]. Acute kidney injury (AKI) is a major clinical problem, affecting up to 5% of all hospitalized patients with acute illness, thus having a great impact on public health resources [2]. AKI is traditionally defined by a rapid decline of renal function, which clinically manifests as an increase of urea and creatinine in serum, associated with disruption of salt and water homeostasis. More importantly, about 8% to 16% of patients with AKI progress to chronic renal failure [3]. There is evidence that even a single episode of AKI predisposes the kidney to maladaptive response to injury leading to progressive loss of function and the development of chronic kidney disease (CKD) [4,5]. In parallel, the incidence of CKD has increased, mainly due to the enhanced prevalence of diabetes and obesity [6]. The current therapies for CKD concentrate on slowing disease progression and, despite beneficial effects, are not sufficient to counteract the disease evolution. A large proportion of patients with end-stage renal disease undergo hemodialysis and/or renal replacement therapy, the latter option with high costs and significant limitation in organ availability [7,8]. Finding new therapeutic strategies for AKI and CKD remains an ongoing quest. In the last decades, innovative stem cell therapies have been tested both as preclinical development and in pilot clinical trials, demonstrating the efficacy of these novel approaches [1,5,9]. More recently, extracellular vesicles (EVs), bioproducts released physiologically from almost all cells, have generated great interest in Regenerative Medicine [10,11,12,13]. This review focuses on stem cell-derived EVs as a new therapeutic option for renal injury repair, with the main focus on mesenchymal stromal cell-derived EVs (MSC-EVs) from different organs.

## 2. Extracellular Vesicles

In the last decade, many studies have characterized new mechanisms of cell-to-cell communication, capable of influencing the phenotype of target cells through the release of bioactive factors [14]. Among all soluble mediators of paracrine communication, EVs possess a central role in both physiological and pathological conditions [15]. EVs are membranous vesicles released by cells of prokaryotic, eukaryotic, and plants, in an evolutionarily conserved manner. Vesicles are heterogeneous in size, sedimentation rate, flotation density, and composition [14,16]. The importance of EVs involves their ability to transfer biologically active molecules and genetic information to other target cells, influencing their function. The first study on EVs appeared many years ago, thanks to Chargaff and West [17], focusing on blood debris. Afterwards, many groups discussed the possibility to consider EVs as cellular discards or bioactive vesicles. It is now well established that EVs interact with cells, inducing target cell stimulation directly or by transferring bioactive molecules [18,19,20]. One of the most significant advances in the role of EVs emerged when EVs were shown to shuttle selected pattern of RNAs transferred to recipient cells and were able to modulate their protein expression pattern [19,21,22]. EVs can be isolated not only from most of the cell types but also from the majority of biological fluids, such as saliva, urine, nasal and bronchial lavage fluids, amniotic and seminal fluids, breast milk, plasma, and serum [23]. In 2011, to confirm the central role of EVs in the regulation of biological processes, the International Society for Extracellular Vesicles (ISEV) was instituted to unify nomenclatures and methodologies for EV isolation and characterization [24,25,26,27].

### EV Composition and Biogenesis

As described in the previous chapter, EVs are very heterogeneous and based on their origin and size; we can distinguish small-size EVs, medium-, and/or large-size EVs [28].

Small-size EVs, previously called exosomes, are vesicles between 30 to 100 nm. They derive from the multivesicular bodies by fusing with the endosomal membranes and are released into the extracellular space [29]. Medium- and/or large-size EVs, also known as microvesicles/ectosomes, are between 50 to 1000 nm. This size range includes different populations of vesicles released by healthy cells up to 200 nm and larger pre-apoptotic bodies. Medium- and/or large-size EVs develop by budding of the plasma membrane [30]. Finally, apoptotic bodies are large-size vesicles from 1 mm up to 5 mm and are shed from the blebbing of the plasma membrane of apoptotic cells [31].

EVs express surface markers specific to their cellular origin and secretion mechanisms. Markers can be distinctive for one group or common for all of them. For example, tetraspanins such as CD9, CD81, and CD63 proteins involved in membrane curvature, are particular to small-size EVs [28]. Moreover, small-size EVs are characterized by the presence of proteins involved in biogenesis, such as Rab, GTPase, annexin, flotillin, components of the endosomal sorting complex required for transport (ESCRT), auxiliary proteins, (ALIX, TSG101, VPS4) and heat shock proteins (HSP70 and HSP90). Medium- and/or large-size EVs express CD40 ligand [29,32] and Annexin A1 [33], while Annexin V is specific for apoptotic bodies [33,34]. Besides, all EV types contain different forms of lipids, such as cholesterols, diglycerides, sphingolipids (including sphingomyelin and ceramide), phospholipids, and glycerophospholipids, fundamental for EV structure [35]. Various types of genetic materials are present within EVs such as noncoding RNAs, mRNAs, miRNAs, and DNAs, each one able to regulate target gene expression at the posttranscriptional level [36]. The expression of miRNAs within EVs, compared with that of originating cells, can be significantly different, suggesting an active and still partially unknown compartmentalization process [33]. The miRNA content exhibits an important role in the biological function of EVs; in fact, it has been shown that they may modulate cell cycle, apoptosis, migration, inflammation, and angiogenesis [37,38].

## 3. MSC-EVs and Tissue Regeneration

The growing evidence in Regenerative Medicine supports the hypothesis that stem cells exert their therapeutic effect by a paracrine/endocrine manner rather than a direct repopulation of the injured tissues [39,40,41,42]. This postulate was strongly supported by numerous in vivo studies demonstrating that the therapeutic benefit of stem cells is orchestrated by their secretome, composed by growth factors, cytokines, chemokines, and EVs [13]. In particular, regarding renal regeneration, Bi et al. [43] showed that the injection of conditioned media from MSCs limits apoptosis and enhances proliferation of tubular cells after a toxic injury, thus promoting kidney repair. The use of EVs, and in particular stem cell-derived EVs, has been proposed as an alternative to stem cell therapy for the regeneration of several injured organs [9,41,44]. MSC-EVs may be isolated from MSCs of different adult tissues such as bone marrow, adipose tissue, peripheral blood, and neonatal birth-associated tissues including placenta, umbilical cord, and cord blood [45]. They are characterized by the expression of the typical mesenchymal stromal markers which include CD44, CD73, CD90, CD105, and CD146 [46].

Moreover, the use of EVs presents many advantages compared with their originating cells, like higher safety profile, lower immunogenicity, and the unfeasibility to maldifferentiate [47,48,49]. They display excellent biological tolerance, an important requirement for therapeutic applications [50]. In addition, EVs possess unique targeting and delivering features as they may be rapidly internalized into target cells [51].

## 4. MSC-EVs and Acute Kidney Injury

The regenerative capacity of EVs is sustained by a high number of publications, and several pre-clinical studies demonstrate that stem cell-derived EVs promote tissue repair and reduce inflammation in different AKI models (Table 1) [52]. The hallmark of AKI is the rapid reduction of renal function in parallel with tubular cell loss, resulting in increased blood urea nitrogen (BUN) and plasma creatinine [53]. In 2009, Bruno et al. [54] demonstrated that the effect of bone marrow (BM) MSC-EVs was superimposable to the one of the originating cells in a model of AKI induced by glycerol injection. BM MSC-EVs accelerate the recovery of injured tubular cells, promoting cell proliferation and protecting cells from apoptosis (Figure 1) [9]. Since that work, many studies have been conducted to confirm the beneficial effect of EVs in several AKI models, and related mechanisms have been explored. At present, it is well recognized that EV activity mainly involves the horizontal transfer of genetic materials [54,55,56,57]. BM MSC-EVs carry specific mRNAs that, in turn, stimulate recipient injured cells for re-entry into the cell cycle [54]. Another group demonstrated that the transfer of human IGF-1 receptor mRNA, present in BM MSC-EVs, to tubular cells is fundamental to trigger renal recovery [58].

Moreover, it has been demonstrated that Drosha-knockdown MSCs release ineffective EVs, tested in in vivo AKI model, sustaining a central role of miRNA cargo [57]. MSC-EVs isolated from bone marrow cells were also tested in toxic AKI models, induced by cisplatin and gentamycin [59,60]. In all toxic models, BM MSC-EVs ameliorated renal function and reduced the classical histological lesions of the disease [4]. The same EV source resulted in an effective ischemia/reperfusion injury (IRI) model that mimics hypoxic insult, a common feature during AKI [61,62]. The effect of MSC-EVs isolated from other tissues was also tested in several AKI models. Similar positive results were obtained using cord blood MSC-EVs that promoted tubular cells dedifferentiation and growth, and Warton Jelly MSC-EVs, that stimulated proliferation and reduced inflammation and apoptosis via mitochondrial protection [63,64,65,66] (Figure 1). In addition, EVs obtained from glomerular MSCs and liver MSCs, human liver stem cells (HLSCs), resulted in protection from AKI [67,68,69]. Altogether these data indicate that MSC-EVs, isolated from different sources, are effective in the amelioration of preclinical models of AKI, targeting multiple aspects of the disease, stimulating cell proliferation, and reducing apoptosis, inflammation, and oxidation [4,9] (Figure 1).

## 5. Conditioning of the Kidney Transplant

Renal transplantation is significantly improving the quality of life of patients with end-stage renal disease; however, chronic allograft nephropathy limits the organ survival and more than one transplant might be required during patient life. The uses of MSCs and MSC-EVs are tested in various clinical protocols related to transplantation, to favor tolerance and to prolong allograft survival [70]. The preconditioning of a kidney with MSCs and MSC-EVs may be another interesting option to limit tissue damage due to ischemia-reperfusion injury and chronic allograft nephropathy. MSCs and MSC-EVs were tested in a rat model of kidney donation after cardiac death (DCD). DCD kidneys treated with MSC-EVs during organ cold perfusion (4 h), showed significantly lower signs of renal damage [71]. In addition, treated kidneys increased energy consumption with up-regulation of enzymes involved in energy metabolism [71]. This approach is gaining increased interest for the pre-transplant graft perfusion in several organs, as it appears to be able to abrogate or strongly reduce ischemic injury.

## 6. MSC-EVs and Chronic Kidney Disease

Several preclinical models are available to mimic the broad range of pathologies defined as CKD. The severity of CKD can manifest itself over time depending on numerous causes. One of the trigger causes is diabetes [72]. Hyperglycemia induces a cascade of events resulting in glomerular and tubule-interstitial fibrosis, with podocyte damage/loss and mesangial cell hypertrophy, a hallmark of diabetic nephropathy. The progression of fibrosis is the leading cause of renal dysfunction not only for diabetic nephropathy but also for other CKDs [73,74]. In this scenario, several groups tested in animal models of CKD, different doses, number, and timing of EV administration, with the intent to set optimal EV regimen (Table 1). EVs isolated from urinary MSCs have been described as effective in the prevention of CKD progression by inhibiting apoptosis in a rat model of diabetic nephropathy induced by streptozotocin injection [75]. EVs induce a reduction of urine volume and apoptosis of podocyte and tubular epithelial cells (Figure 1). Urinary MSC-EVs carry transforming growth factor-β1, angiogenin, and bone morphogenetic protein-7, drivers of the observed reno-protective activity [75]. In addition, the direct administration of MSC exosomes under the renal capsule generated a rapid improvement of renal morphology, demonstrated in the same animal model [76]. Recently, EVs isolated from BM MSCs and from liver MSCs have been shown to be effective in the reversion of renal fibrosis in an already established diabetic nephropathy model [8]. MSC-EVs and HLSC-EVs contain a selection of antifibrotic miRNAs able to downregulate profibrotic genes, restoring normal renal function [8]. Similar positive results were obtained by multiple injections of HLSC-EVs in a CKD model induced by aristolochic acid [77].

Other in vivo models of CKD are the surgical five-sixth resection of the kidney tissue and the obstruction of the ureter, leading to glomerulosclerosis and fibrosis [78]. In both CKD models, multiple injections of BM MSC-EVs prevented renal failure [79,80]. In a similar model, combined with diet, multiple administrations of a conditioned medium, purified from human embryonic MSCs, slowed the deterioration of renal function [81]. Moreover, in a porcine model of metabolic syndrome and renal artery stenosis, a single intrarenal administration of adipose tissue-derived MSC-EVs reduced renal inflammation and fibrosis by delivery of IL10 [82].

The robustness of preclinical data about the therapeutic efficacy of MSC-EVs in acute and chronic models is encouraging to go further towards clinical studies.

## 7. MSC-EVs and Clinical Trials

The translation of EV-based therapy into clinical practice requires the clarification of several critical issues [13]. The major one to be considered is the identification of optimal protocols for EV production, isolation, and storage [13]. Similarly, the determination of potency assays to test the efficacy of each EV batch is mandatory. In fact, the majority of approved clinical trials implying EVs (listed in www.clinicaltrials.gov) focus on diagnostic purposes. However, at present, there are four clinical trials involving MSC-EVs for therapeutic use (Table 2). Two of them are designed by Nassar et al. [83] at the Sahel Teaching Hospital of the University of Cairo. Both trials imply the use of EVs isolated from cord blood MSCs [83]. The first study aims to evaluate the effect of consecutive doses of MSC-EVs in 20 patients with type 1 diabetes, with a follow up of three months [13]. The results are not available yet. The second study enrolled 20 patients with CKD and results are already published [83]. The authors observed an improvement of renal function with amelioration of glomerular filtration, proteinuria, and BUN in patients one year after EV administration (two doses). Moreover, EVs displayed an anti-inflammatory activity, decreasing TNF-α and increasing IL-10. The results of this clinical study are promising in terms of feasibility and efficacy for MSC-EV therapeutic use. Another potential application in which preclinical studies are robust and convincing is the use of MSC-EVs to promote macular regeneration. There is an ongoing clinical trial in China focusing on the safety and efficacy of exosomes isolated from cord tissue-derived MSCs in patients with refractory macular holes in the eye. Finally, a clinical trial, which involves the injection of MSC-EVs engineered with miR-124 for the treatment of patients after acute ischemic stroke, was approved in Iran,

The number of clinical trials on EVs as a therapeutic strategy will increase enormously in the next years and, hopefully, their use will enter into clinical practice.

## 8. EV Engineering and Future Strategies

In the constant quest to broaden the therapeutic applications of EVs, further approaches focused on the enhancement of EV efficacy by engineering. The natural origin of EVs, along with their spheroid shape and cargo ability, makes them ideal candidates for the efficient loading of therapeutic molecules [84]. The strategy to engineer EVs with pro-regenerative molecules or specific drugs is currently gaining an increasing interest [85]. EVs may be engineered to potentiate their therapeutic cargo by increasing the levels of active molecules (proteins or RNAs) already present within EVs or to modify their biodistribution/stability by changing the composition of surface molecules. The strategy to deliver therapeutic RNAs possesses an excellent potential and a wide range of applicability; however, the polar RNA molecules are exposed to rapid digestion by extracellular RNases [50,86]. The use of synthetic nanoparticles has also been explored with some limitations [87,88]. For these reasons, EVs are the central point of intense research [50]. At present, the existing methods for EV engineering are divided into two categories: Direct and indirect methods, indicating the direct modification of the EVs or the engineering of the cell of origin used for EV production.

Direct EV engineering can be done with multiple techniques: Incubation, electroporation, sonication, freeze/thaw cycles, and saponin-assisted method without significantly impairing EV constitution and functionality (Figure 2) [89]. The incubation is a passive method preferred for the loading of hydrophobic compounds, with a higher efficiency compared to those obtained with liposomes. The reason may be the presence of particular domains within EV membranes, absent in artificial membranes of liposomes [90]. Sun et al. [91], for example, mixed purified EVs with curcumin, a natural compound with antioxidant and anti-inflammatory activities, and they demonstrated an increased efficacy compared with those of naive EVs when injected into mice with septic shock.

Exogenous genetic material (small RNAs or miRNAs) is generally added to EVs using electroporation, as they are hydrophilic molecules. To define the most efficient protocol, Pomatto et al. [92] tested different voltages and number of pulses and described 750 V and 10 pulses as the optimal one, with the highest RNA loading without significant EV damage [92].

As an example, a non-coding RNA (Lnc-RNA-H19) has been transfected into high-yield nano-EVs to create an effective drug delivery system for wound healing in diabetics [93]. In addition, sonication or cycles of a deep freeze and then slow thaw are two alternative methods for inserting different molecules into isolated EVs [94,95]. Moreover, it has also been shown that the saponin-assisted encapsulation method allows the highest loading efficacy and protection versus protease degradation [89]. All options mentioned above can be combined to improve final loading efficacy. The modification of the genome by CRISPR/Cas technology, which alone has low efficacy of delivery, is a new potential tool to be inserted into EVs by electroporation [96].

The second category of engineering technology is based on the modification of EV originating cells that allows subsequent isolation of EVs, which already express the desired molecule. For example, it has been demonstrated in an in vivo model of unilateral ureteral obstruction that MSCs engineered to overexpress miRlet7c selectively localize into the injured kidney and upregulate miR-let7c expression, attenuating kidney injury. Similarly, exosomes derived from engineered MSCs were able to selectively transfer miR-let7c to damaged kidney cells resulting in antifibrotic functions [97]. In a similar approach, MSCs were engineered to overexpress pro-regenerative miRNAs, such miR10a, miR127, and miR486, and deriving EVs were tested in models of acute renal injury [98]. EVs obtained from engineered MSCs were more effective than EVs derived from naïve MSCs when used at low doses [98].

## 9. Conclusions

The number of studies on the use of EVs, especially those derived from MSCs, for the treatment of AKI and CKD is continuously increasing, and EVs are considered a promising approach for tissue regeneration. The pro-regenerative effect of EVs is now well established for AKI, sustained by convincing results in a large number of different experimental models. The regenerating role of MSC-EVs in the slowdown of CKD, at variance, is still limited to a restricted number of preclinical models and should be better investigated. The translation of this approach for clinical use, based on ongoing and future clinical trials, will open a new scenario in Regenerative Medicine. Finally, EVs could be further exploited as a carrier for the delivery of exogenous materials such as RNAs, proteins or existing small drugs. An accurate setting of therapeutic doses and schedule are still needed.

## Figures and Tables

**Figure 1 cells-08-01240-f001:**
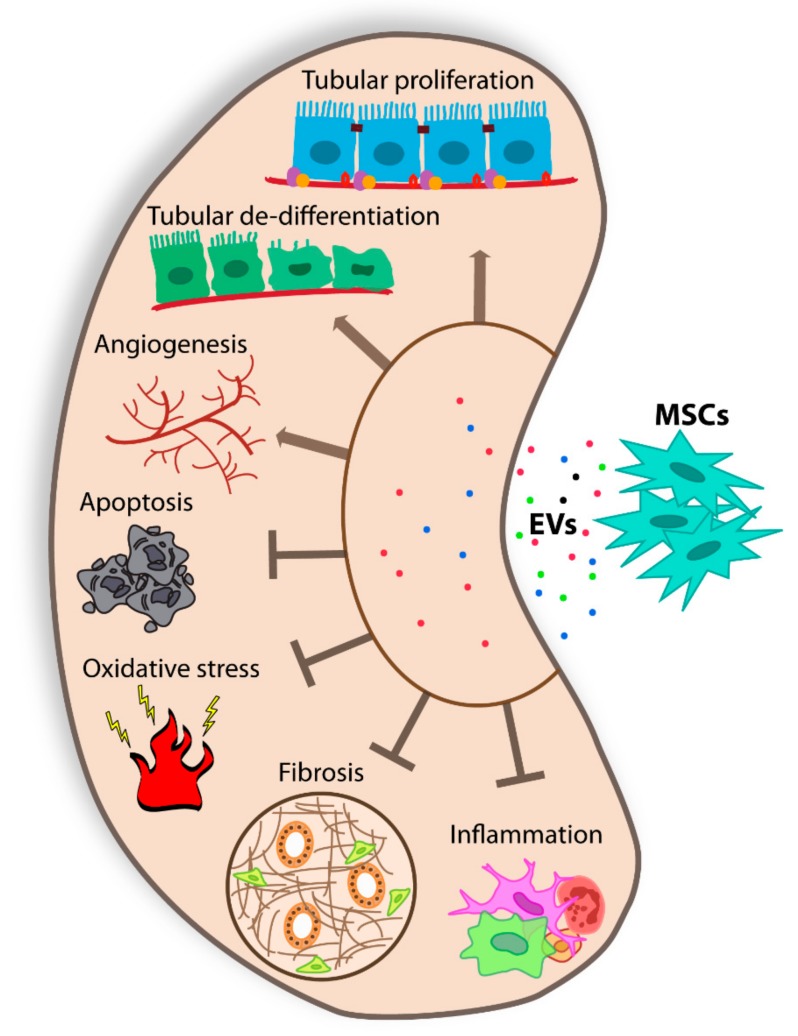
Schematic representation of the effects of MSC-EVs on renal injury.

**Figure 2 cells-08-01240-f002:**
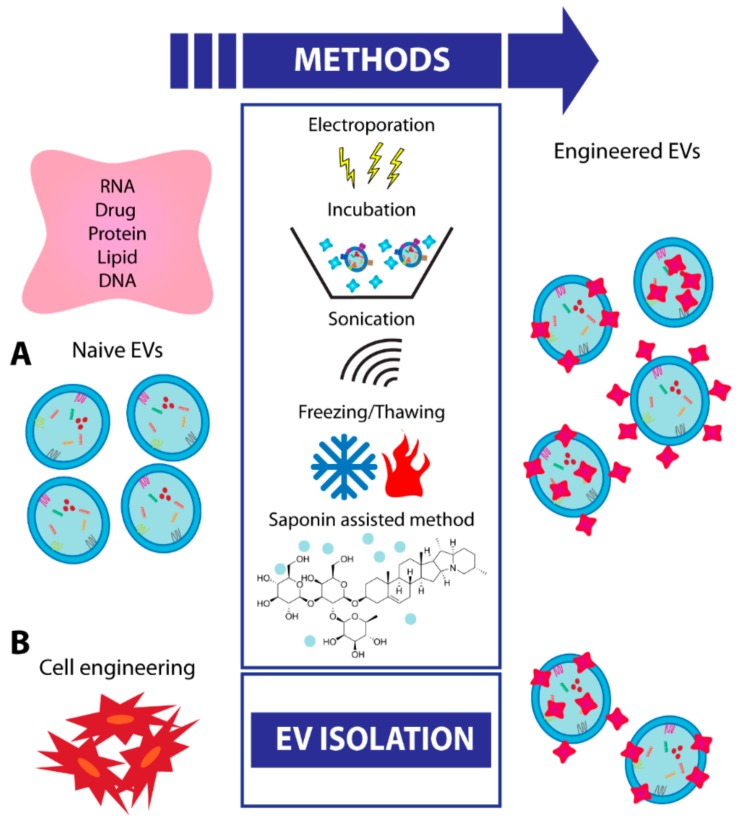
Schematic representation of different procedures for EV engineering. (**A**) Schematic representation of techniques for engineering EVs after their isolation (direct method). (**B**) Schematic representation of cell engineering followed by EV isolation (indirect method).

**Table 1 cells-08-01240-t001:** Mesenchymal stromal cell-extracellular vesicle (MSC-EV) administration in animal models of renal damage. EVs released by MSCs derived from different tissues are effective in models of acute kidney injury (AKI) and chronic kidney disease (CKD). EV sources, animal models, doses, and route of administration are listed. Abbreviation: IRI Ischemia Reperfusion Injury.

MSC Origin	In vivo Models	Type of Injury	Injection	Administration	References
Bone Marrow	Glycerol	AKI	Single: 15 μg Single: 2.2 × 10^8^	Intravenously	Bruno et al. [54] Collino et al. [57]
	IRI	AKI	Single: 30 μg	Intravenously	Gatti et al. [61]
	Cisplatin	AKI	Single: 100 μg	Intravenously	Bruno et al. [59]
	Gentamicin	AKI	Multiple: 100 μg	Intravenously	Reis et al. [60]
	IRI	AKI	Single:200 μg	Into renal capsule	Shen B et al. [62]
	IRI	CKD	Single: 30 μg	Intravenously	Gatti et al. [61]
	Cisplatin	CKD	Multiple: 100 μg followed by 50 μg every 4 days	Intravenously	Bruno et al. [59]
	Remnant kidney	CKD	Single: 30 μg	Caudal vein	He et al. [79]
	Type 1 diabetes	CKD	Single: 5.3 × 10 exosomes	Renal subcapsular	Nagaishi et al. [76]
	Unilateral ureteral obstruction	CKD	Single: 30 μg	Caudal vein	He et al. [80]
	Type 1 diabetes	CKD	Multiple: 1 × 10^10^/dose	Intravenously	Grange et al. [8]
Cord blood	Cisplatin	AKI	Single: 200 μg	Caudal vein	Zhou et al. [63]
	IRI	AKI	Single: 30 μg	Caudal vein	Ju et al. [65]
Warton Jelly	IRI	AKI	Single:100 μg	Caudal vein Caudal vein	Zou et al. [64] Gu et al. [66]
Renal	IRI	AKI	Single: 2 × 10^7^	Intravenously	Choi et al. [68]
	IRI	AKI	Single: 400 × 10^6^	Intravenously	Ranghino et al. [67]
Liver	Glycerol	AKI	Single:1.88 ± 0.6 × 10^9^ Single: 5.53 ± 2.1 × 10^9^	Intravenously Intravenously	Herrera Sanchez et al. [69]
	Aristolochic acid nephropathy	CKD	Multiple	Intravenously	Kholia et al. [77]
	Type 1 diabetes	CKD	Multiple: 1 × 10^10^/dose	Intravenously	Grange et al. [9]
Urine	Type 1 diabetes	CKD	Multiple: 100 μg weekly 12 times	Intravenously	Jiang et al. [75]
Embryonic	Remnant kidney and specic diet L-N^G^–nitroarginine and 6% NaCl	CKD	Multiple: 7 μg twice daily for 4 consecutive days	Intravenously	Van Koppen et al. [81]
**Adipose tissue**	Porcine model of metabolic syndrome and renal artery stenosis	CKD	Single: 1 × 10^10^	Intra renal	Eirin et al. [82]

**Table 2 cells-08-01240-t002:** Clinical trials using MSC-EVs for therapeutic purposes. Application, dose, number of patients, and follow-up are listed. Moreover, identification number and state of trial are reported.

Disease	Intervention	N. Pats	Follow Up	State	Location	Number/Ref.
Diabetes Mellitus Type 1	Two doses of MSC-EVs	20	3 months	Unknown	Sahel Teaching Hospital Sahel, Cairo, Egypt	NCT02138331
Chronic kidney disease	Two doses of umbilical cord MSC-EVs (100 μg/kg/dose)	20	1 year	Concluded	Sahel Teaching Hospital Sahel, Cairo, Egypt	Nassar et al. [83]
Macular degeneration	20–50 mg of cord tissue MSC-EVs injected directly around macular hole	44	24 weeks	Recruiting	Tianjin Medical University Hospital Tianjin, China	NCT03437759
Cerebrovascular disorders acute ischemic stroke	Allogenic MSC-EVs enriched by miR-124	5	12 months	Not yet recruiting	Shahid Beheshti University of Medical Sciences, Teheran Iran	NCT03384433

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
