# Peer review of "Stem Cell-Derived Extracellular Vesicles and Kidney Regeneration"

_cells, 2019, doi:10.3390/cells8101240_

Round 1

Reviewer 1 Report

The review is well written and it is correctly organized.

I only have some minor remarks.

In the abstract, last line, what is the new strategy to broaden EV efficacy? If I understood correctly, the authors describe multiple strategy for engineering EVs.

Figure 2. As the caption states that the figure is a representation of different procedures for EV engineering, I believe the authors should add a scheme describing also the second category of engineering technology based on the modification of EV originating cells.

Page 2 line 67. Is the title “Ev composition and biogenesis” supposed to be like that with no number (i.e. 2.1)?

Author Response

We thank the reviewer for the positive comments and suggestions, that we took into consideration, as described below:

In the abstract, last line, what is the new strategy to broaden EV efficacy? If I understood correctly, the authors describe multiple strategy for engineering EVs.

Answer. Thank you for this suggestion. We rephrase the sentence. (See Abstract, last line).

Figure 2. As the caption states that the figure is a representation of different procedures for EV engineering, I believe the authors should add a scheme describing also the second category of engineering technology based on the modification of EV originating cells.

Answer. Thank you for this suggestion. We change the figure adding the second category of engineering technology. (See Figure 2).

Page 2 line 67. Is the title “Ev composition and biogenesis” supposed to be like that with no number (i.e. 2.1)?

Answer. Yes, there is no number because we did not classify chapter and sub-chapter by number. This is the only sub-chapter and we hope the editorial processing will deal with that.

Reviewer 2 Report

The authors present a thorough and extensive review of the use of extracellular vesicles for the treatment of kidney disease. Overall, the manuscript is logically laid out and the topic is thoroughly discussed in a way that is easy for the reader for follow and understand. There are a couple of minor spelling and grammatical errors that need to be corrected throughout the manuscript but overall it is well written. Other than the spelling and grammar corrections I have no further comments.

Author Response

The authors present a thorough and extensive review of the use of extracellular vesicles for the treatment of kidney disease. Overall, the manuscript is logically laid out and the topic is thoroughly discussed in a way that is easy for the reader for follow and understand. There are a couple of minor spelling and grammatical errors that need to be corrected throughout the manuscript but overall it is well written. Other than the spelling and grammar corrections I have no further comments.

Answer. Thank you for your positive comments. We carefully revised the text.

Reviewer 3 Report

This manuscript represents a comprehensive review of literature in the field of Stem Cell Derived Extracellular Vesicles and Kidney Rigeneration. I found it useful for readers and experts in the field

Minor: for the paragraph entitled "EV composition and biogenesis" I would refer to MISEV2018 guidelines and I would apply nomenclature suggested in those guidelines 

Author Response

To Reviewer 3.

We thank the reviewer for the positive comments and suggestions.

Minor: for the paragraph entitled "EV composition and biogenesis" I would refer to MISEV2018 guidelines and I would apply nomenclature suggested in those guidelines.

Answer. Thank you for your specification; this is an important point. We correct accordingly, indicating small and large size EVs rather than exosomes and ectosomes/microvesicles.